# Therapeutic Strategies to Target Calcium Dysregulation in Alzheimer’s Disease

**DOI:** 10.3390/cells9112513

**Published:** 2020-11-20

**Authors:** Maria Calvo-Rodriguez, Elizabeth K. Kharitonova, Brian J. Bacskai

**Affiliations:** Alzheimer Research Unit, Department of Neurology, Massachusetts General Hospital and Harvard Medical School, Charlestown, MA 02129, USA; mcalvorodriguez@mgh.harvard.edu (M.C.-R.); ekharitonova@mgh.harvard.edu (E.K.K.)

**Keywords:** calcium homeostasis, Alzheimer’s disease, therapeutics, amyloid, tau, endoplasmic reticulum, mitochondria, lysosomes

## Abstract

Alzheimer’s disease (AD) is the most common form of dementia, affecting millions of people worldwide. Unfortunately, none of the current treatments are effective at improving cognitive function in AD patients and, therefore, there is an urgent need for the development of new therapies that target the early cause(s) of AD. Intracellular calcium (Ca^2+^) regulation is critical for proper cellular and neuronal function. It has been suggested that Ca^2+^ dyshomeostasis is an upstream factor of many neurodegenerative diseases, including AD. For this reason, chemical agents or small molecules aimed at targeting or correcting this Ca^2+^ dysregulation might serve as therapeutic strategies to prevent the development of AD. Moreover, neurons are not alone in exhibiting Ca^2+^ dyshomeostasis, since Ca^2+^ disruption is observed in other cell types in the brain in AD. In this review, we examine the distinct Ca^2+^ channels and compartments involved in the disease mechanisms that could be potential targets in AD.

## 1. Calcium Dysregulation Is a Hallmark of Alzheimer’s Disease

Alzheimer’s disease (AD) is the most common form of dementia, affecting more than 30 million people worldwide. It is characterized by accumulation of extracellular amyloid β (Aβ) plaques—or senile plaques—composed of Aβ peptide, intraneuronal fibrillary tangles (NFTs) comprising hyperphosphorylated and misfolded microtubule-associated protein tau, and selective neuronal loss, particularly in brain regions like the neocortex and hippocampus, eventually leading to memory loss and a decline in cognitive function. Most AD cases are sporadic (SAD), with less than 1% due to genetic mutations. Risk factors, such as aging, lifestyle, obesity, or diabetes, or genetic factors such as carrying the allele ε4 in the apolipoprotein E (ApoE) gene predispose individuals to SAD development [1]. Genetically inherited forms of AD (familial AD, FAD) show early onset and are caused by mutations in genes coding for presenilin (PS) 1, PS2, or amyloid precursor protein (APP), all involved in the Aβ generation pathway. Other than the onset, there are no clear differences regarding symptoms or histopathological features between SAD and FAD. Different hypotheses have been proposed to explain the origin of AD. The relation to genetics in FAD supported the “amyloid cascade hypothesis”, which suggests that AD pathogenesis is initiated by overproduction of Aβ and/or failure of its clearance mechanisms, upstream of tau dysregulation [2]. However, other hypotheses that explain the etiology of AD are being considered. The “cholinergic hypothesis” [3], “tau propagation hypothesis” [4], “inflammatory hypothesis” [5], or “glymphatic system hypothesis” [6] are among the most relevant.

Intracellular calcium (Ca^2+^) is an important second messenger that regulates multiple cellular functions, such as synaptic plasticity, action potentials, and learning and memory. Ca^2+^ dyshomeostasis, on the other hand, contributes to detrimental mechanisms such as necrosis, apoptosis, autophagy deficits, and neurodegeneration. Perturbations in intracellular Ca^2+^ are involved in many neurodegenerative diseases including AD, Parkinson’s disease, and Huntington’s disease [7]. Back in the mid-1980s, Khachaturian proposed that Ca^2+^ dysregulation led to neurodegeneration, suggesting that a sustained imbalance of cellular Ca^2+^ could disrupt normal neuronal functions and lead to neurodegenerative diseases such as AD [8]. Since then, many reports have shown Ca^2+^ dysregulation in AD (both in SAD [9,10] and in FAD [11]), animal models of the disorder [12,13,14,15,16,17,18,19], and cells from human AD patients [20]. The “Ca^2+^ hypothesis of Alzheimer’s disease” [21] postulates that activation of the amyloidogenic pathway causes a remodeling of normal neuronal Ca^2+^ signaling pathways, which then alters Ca^2+^ homeostasis and leads to the disruption of the mechanisms involved in learning and memory. Neuronal Ca^2+^ dyshomeostasis seems to manifest early in AD progression prior to the development of histopathological markers or clinical symptoms [22]. Similarly, AD is also marked by Ca^2+^ disruption in other cells in the brain such as astrocytes and microglia. Whether disruption of Ca^2+^ homeostasis is cause or consequence of AD pathology is still a matter of debate.

Up to date, there are only two types of Food and Drug Administration (FDA)-approved therapies for AD treatment (www.alzforum.org)—acetylcholinesterase inhibitors and *N*-methyl-d-aspartate receptor (NMDAR) antagonists—and neither can cure or reverse the disease, but can, at least, transiently relieve patients’ symptoms [23]. Unfortunately, drugs targeting Aβ have been mostly unsuccessful. Although these therapies have shown some success in clearing Aβ plaques from the AD brain, they have failed to relieve the cognitive decline of AD patients in clinical trials [24], with the exception of aducanumab, which demonstrated both clearance of plaques and modest gains in cognitive function [25]. In addition, the well-known lack of correlation between cognitive symptoms and Aβ deposition further supports the idea of the need for different approaches [26]. Ca^2+^ dyshomeostasis is an early molecular defect in AD and might precede Aβ and tau deposition [22]. Therefore, therapeutics that stabilize Ca^2+^ signals may represent an alternative strategy for treating AD. In the remaining sections, we review human data and those generated from experimental models, and we discuss the different strategies for targeting Ca^2+^ dysregulation—including specific Ca^2+^ channels and different cell types—that could be used as therapeutics in AD.

## 2. Neuronal Ca^2+^ as a Therapeutic Target in AD

Ca^2+^ is a fundamental regulator of neuronal fate; thus, intracellular Ca^2+^ homeostasis must be finely tuned in physiological conditions. In the extracellular space, Ca^2+^ concentration is maintained between 1.1 and 1.4 mM, whereas resting cytosolic levels within neurons are maintained in the nM range (50–300 nM) [27]. After cell activation, intracellular Ca^2+^ concentrations increase rapidly to the μM range. This Ca^2+^ gradient allows the initiation of different signaling cascades. Ca^2+^ levels in the endoplasmic reticulum (ER) are nearly a thousand times higher than those of the cytoplasm [28]. Ca^2+^ signals are generated by the influx of Ca^2+^ from the extracellular space or by Ca^2+^ release from intracellular stores. Ca^2+^ enters neurons mainly through plasma membrane channels and is then buffered by Ca^2+^-binding proteins and organelles such as mitochondria. Even though the mechanisms responsible for neuronal Ca^2+^ dysregulation in AD are not completely understood, as discussed in the sections below, evidence shows that different compartments and/or organelles are involved (Figure 1).

### 2.1. Targeting Plasma Membrane Receptors and Cytosolic Ca^2+^

Proper intracellular Ca^2+^ homeostasis is crucial for many neuronal functions. Disruption of this homeostasis might be one of the main mechanisms via which Aβ and tau exert their neurotoxicity. The main plasma membrane channels involved in neuronal Ca^2+^ influx from the extracellular space are voltage-gated Ca^2+^ channels (VGCCs), which allow Ca^2+^ influx following neuronal depolarization, and receptor-operated Ca^2+^ channels (ROCs), which open upon specific binding of the agonist, with NMDARs among the most important examples.

Toxic Aβ increases cytosolic Ca^2+^, which may affect a variety of enzymes (such as proteases or phosphatases), promote cytoskeletal modifications, cause the generation of free radicals, or trigger neuronal apoptosis [29]. It has been proposed that Aβ can overactivate channels and/or form pores in the cytosolic plasma membrane, allowing massive influx of Ca^2+^ from the extracellular space and increasing the overall Ca^2+^ levels in the cytosol, severely limiting normal cellular function [30,31]. Aβ potentiates Ca^2+^ influx through VGCCs, particularly L-type VGCCs. Excessive Ca^2+^ influx through these channels has been observed in cultured neurons following Aβ exposure and was shown to be blocked by the L-type VGCC inhibitor nimodipine [32]. This phenomenon, however, was not observed in brain slices from AD mouse models [33]. In addition, AD patients taking L-type VGCCs inhibitors such as nilvadipine (NILVAD multicenter trial) showed reduced Aβ levels but no improvement in cognitive decline [34,35]. Acute application of tau aggregates has also been observed to increase cytosolic Ca^2+^ and elevate reactive oxygen species (ROS) production via nicotinamide adenine dinucleotide phosphate (NADPH), an effect that can be prevented by nifedipine and verapamil, both L-type VGCC inhibitors [36]. This suggests that tau fibrils could also incorporate into the cell membrane to activate VGCCs and lead to neuronal dysfunction [36].

CALHM1 (Ca^2+^ homeostasis modulator protein 1) is a Ca^2+^ channel highly expressed in neurons in the hippocampus that allows cytosolic Ca^2+^ influx in response to decreases in extracellular Ca^2+^ [37]. Its activation triggers different kinase signaling cascades in neurons. The *CALMH1* polymorphism P86L has been proposed as a risk factor for late-onset SAD [9,37], an argument that has been challenged by other groups [38,39]. Nevertheless, increased levels of Aβ have been observed in transfected cells expressing the P86L polymorphism, suggesting a role for CALMH1 in AD [37]. Additionally, the P86L polymorphism alters the channel permeability to Ca^2+^ [37]. A recent study demonstrated that CALHM1 deficiency in mice leads to cognitive and neuronal deficits, which manifest memory impairment and hippocampal long-term potentiation (LTP) [40], pointing to CALHM1 as a potential treatment target in AD.

NMDARs are a subfamily of ionotropic glutamate receptors involved in the excitatory synaptic transmission and synaptic plasticity of the brain. Specific types of NMDARs are much more permeable to Ca^2+^ than other ionotropic glutamate receptors and are often implicated in neuronal pathophysiology. NMDARs are mainly composed of GluN2A and GluN2B in the brain areas most affected in AD [41]. Extra-synaptic GluN2B-containing NMDARs have been associated with excitotoxicity (the excessive neuronal death induced by cellular Ca^2+^ overload due to excessive stimulation of glutamate receptors) and the toxic effect of Aβ oligomers in AD [42,43]. For this reason, selective GluN2B subunit antagonists may be a strategy to prevent synaptic dysfunction in AD. Aβ_42_ peptides interact with NMDARs, potentiating their activity and leading to increased Ca^2+^ influx, thus contributing to the synapse loss observed in AD [44,45]. Additionally, as demonstrated in mouse models of AD, glutamate-induced excitotoxicity is inhibited by tau reduction [46] and exacerbated by tau overexpression [47,48]. In turn, glutamate-induced excitotoxicity increases tau expression [49] and phosphorylation [50], while activation of extra-synaptic NMDAR leads to tau overexpression, neuronal degeneration, and cell loss [51]. Memantine—a weak NMDAR antagonist—is one of the two FDA-approved drugs to treat AD patients and the only NMDAR antagonist [23]. It provides modest improvements to memory and cognitive performance in moderate to severe AD patients [52,53]. Memantine restricts excessive Ca^2+^ influx, thus reducing neuronal excitotoxicity, and, due to its low activity, the basal NMDAR function is preserved. Memantine has also shown neuroprotective effects against oxidative stress, neuroinflammation, and tau phosphorylation [54,55].

Ionotropic neuronal nicotinic acetylcholine receptors (nAChRs) respond to the neurotransmitter acetylcholine (ACh) and to drugs such as the agonist nicotine. They are permeable to Na^+^, K^+^, and Ca^2+^. The nAChRs expressing the α7 subunit have the highest conductance for Ca^2+^ and are found in brain regions most susceptible to AD [56]. In the basal forebrain, cholinergic neuronal loss and decreased levels of ACh mediate cholinergic impairment, which eventually leads to short-term memory loss [57,58,59]. The loss of cholinergic innervation in early AD led to the “cholinergic hypothesis of AD” [3]. Galantamine and rivastigmine (for use in mild to moderate AD) and donepezil (in mild to severe AD) are the cholinesterase inhibitors FDA-approved to treat AD [23]. These drugs act by increasing ACh levels, which delay the progression of AD through Ca^2+^-dependent mechanisms. Furthermore, supplemented with memantine, it has been proposed that this combination could provide greater benefits on behavior, cognition, and global outcomes in AD [60].

Exposure of hippocampal and cortical neurons to tau also increased intracellular Ca^2+^ levels through muscarinic receptors [61]. Interestingly, Ca^2+^ activates many kinases, including those responsible for tau phosphorylation—such as glycogen synthase kinase 3β (GSK3β)—and, therefore, Ca^2+^ dyshomeostasis may increase tau phosphorylation and NFT formation [62]. Given that tau pathology correlates better with cognitive impairments than Aβ deposition, tau targeting is expected to be more effective once clinical symptoms emerge [63]. It has long been known that tau-expressing cells secrete normal and pathological tau [64], which can be taken up by other cells, seeding and spreading tau pathology [4,65,66,67]. Led by immunotherapy approaches, the efforts to target tau with therapeutics focus on reducing tau pathology by limiting the spread of extracellular tau across brain regions [68]. Anti-tau immunotherapy has shown potential in numerous clinical studies. Both active and passive tau immunization seem to offer a promising option by reducing tau pathology [69]. Active tau immunization, however, seems to elicit a risk of adverse immune reactions from targeting the normal protein. Other tested approaches involve reducing tau expression (with small interfering RNA or antisense oligonucleotides; siRNA and ASOs, respectively), targeting tau modifications, reducing tau aggregation, and stabilizing microtubules. Preventing or reducing pathologic tau has been shown to improve cognitive and motor impairments in animal models with neurofibrillary pathology, and several tau antibodies and vaccines have been tested in preclinical studies in the last years. Immunotherapy is currently at the stage of drug development (recently reviewed in [68,69,70]), and, as of today, eight humanized tau antibodies and two tau vaccines are under clinical trial for AD or frontotemporal dementia [71] (www.alzforum.org).

The use of intravital imaging and transgenic mouse models of AD have allowed for direct observation of cytosolic Ca^2+^ dysregulation. In vivo, neuronal cytosolic Ca^2+^ dyshomeostasis is more likely to be observed in the vicinity of amyloid β plaques, but is detectable in neurons throughout the cortex [15]. Higher Ca^2+^ levels were observed in neurons close to amyloid plaques in a commonly used mouse model of cerebral amyloidosis (APP_Swe_xPS1ΔE9, APP/PS1) [15], but only after plaque deposition and not before. Cytosolic Ca^2+^ overload was absent in mice harboring only the PS1 mutation (typically lacking plaque deposition). The mechanisms of Ca^2+^ dysregulation involved activation of calcineurin (CaN), a Ca^2+^/calmodulin-dependent protein phosphatase sensitive to subtle rises in intracellular Ca^2+^ levels, and whose activation induces long-term depression (LTD). Ca^2+^ dysregulation in neurites was linked to neurodegeneration (neuritic blebbing and beading), which can be partially prevented by inhibiting CaN [15]. Elevated Ca^2+^ levels in the neurites impair synapses, by increasing the frequency of spontaneous synaptic potentials and reducing plasticity. In addition, pathological increases in neuronal network activity—observed as increased frequencies of somatic Ca^2+^ transients—potentiate Aβ release into the extracellular space [72,73]. In the APP23/PS45 mouse model of AD (overexpressing mutant APP_Swe_ and mutant PS1_G348A_), neuronal hyperactivity was observed around amyloid plaques in the cortex, only after plaque deposition [14]. Hyperactive neurons, however, were found in the CA1 region of the hippocampus in pre-depositing animals [13]. Direct application of soluble Aβ onto the wild-type (Wt) naïve brain increased cytosolic Ca^2+^ levels [12] and induced neuronal hyperactivity [13]. Acute treatment with the γ-secretase inhibitor LY-411575, which reduces soluble Aβ levels, normalized the frequency of Ca^2+^ transients prior to plaque deposition [13].

Interestingly, AD patients are more prone to developing epileptic seizures [74]. Blocking network hyperactivity with the antiepileptic drug levetiracetam improves learning and memory, reverses behavioral abnormalities, and reverts synaptic deficits in the hippocampus in an AD mouse model [75]. In the same way, it has been observed that tau is implicated in neuronal circuit deficits in mouse models of AD expressing both Aβ and tau. Tau effects dominate those of Aβ and are mostly dependent on the presence of soluble tau [16]. According to the authors, this dramatic effect could suggest a possible cellular explanation contributing to disappointing results of anti-Aβ therapeutic trials. This abnormal network activity and its resultant AD-related cognitive deficits in mice point to neuronal hyperactivity as a promising therapeutic target in AD.

Aducanumab is a high-affinity, fully human immunoglobulin G1 (IgG1) monoclonal antibody that selectively binds to aggregated Aβ fibrils and soluble oligomers (and not monomers) in the brain parenchyma [25]. It was shown that it could ameliorate Ca^2+^ dysregulation in AD. Using multiphoton microscopy and a Ca^2+^ reporter, it was observed that a single topical application of the antibody onto the brain surface of mice depositing amyloid plaques (Tg2576 AD model) led to a reduction in existing amyloid deposits [76]. Peripheral administration of the antibody over a period of 6 months rescued Ca^2+^ overload in transgenic neurites, restoring them to control levels within 2 weeks. The authors suggested that aducanumab exerted its function by targeting amyloid deposits, including soluble oligomeric Aβ [76]. In March 2019, the termination of all aducanumab clinical trials was announced after an interim analysis of EMERGE and ENGAGE trials predicted the phase III placebo-controlled studies would not meet their primary end points. However, in a subsequent analysis of a larger dataset from the EMERGE trial, aducanumab met the primary end point, and the FDA accepted the aducanumab application for review [77]. If the case is approved, aducanumab would be the first drug to combat the root causes of AD.

### 2.2. Targeting ER Ca^2+^ and SOCE

The ER is an important subcellular organelle involved in protein synthesis, modification, and folding. Additionally, it is a dominant Ca^2+^ reservoir in the cell, critical for maintaining intracellular Ca^2+^ levels [27]. Ca^2+^ is released from the ER after activation of either inositol 1,4,5-trisphosphate receptors (IP_3_Rs) or ryanodine receptors (RyRs). Ca^2+^ efflux from the ER modulates a range of neuronal processes, including regulation of axodendritic growth and morphology or synaptic vesicle release [78]. The sarco-endoplasmic reticulum ATPase (SERCA) pump, which actively consumes ATP, is important for extruding Ca^2+^ into the ER lumen, where it is sequestered by binding to proteins such as calsequestrin and calretinin, priming this organelle as a critical component of Ca^2+^ buffering.

Impaired IP_3_R signaling in the ER was an early discovery in AD. It was shown that human cells from FAD patients exhibited enhanced Ca^2+^ release in response to IP_3_R-generating stimuli [79]. Fibroblasts from asymptomatic members of AD families [80], as well as PS1 knock-in mice and other presymptomatic AD mouse models, showed the same enhancement [81]. These observations suggested that FAD mutations contribute to Ca^2+^ dysregulation, even before pathology deposition or cognitive impairments were evident. A reduction in IP_3_R expression can normalize Ca^2+^ homeostasis and restore hippocampal LTP in mouse models of AD [82]. PSs are transmembrane proteins found in the ER membranes and form the catalytic core of the γ-secretase complex that processes APP and other type 1 transmembrane proteins, such as Notch [83,84]. PSs are essential for learning and memory, as well as neuronal survival during aging in the murine cerebral cortex [85,86]. Mutations in PSs have been shown to affect APP processing, leading to increased production of the more hydrophobic neurotoxic form Aβ_42_ [87,88,89] or increasing the Aβ42/40 production ratio [90]. It has also been proposed that *PSEN* mutations cause a loss of presenilin function in the brain, triggering neurodegeneration and dementia in FAD [91]. Mutations in PS1 and PS2 might stimulate IP_3_Rs, leading to exaggerated Ca^2+^ release through these channels [79,80,81]. An alternative hypothesis suggested that PSs function as passive low-conductance leak channels in the ER membrane. AD-associated PS mutations might impair this leak function, resulting in ER Ca^2+^ overload [92] and leading to exaggerated increases in cytosolic Ca^2+^ upon stimulation of Ca^2+^ release. However, these observations have not been supported by other groups [93,94], and, despite extensive research, this subject is still a matter of controversy. Recently, it was proposed that the ER-based transmembrane and coiled-coil domain TMCO1 could be responsible for the ER Ca^2+^ leak [95].

RyR Ca^2+^ dysregulation was also observed before the histopathology and cognitive decline in AD. Both human brain tissue from AD patients and AD mouse models have shown increased expression of RyR (particularly RyR_2_) in affected brain regions in AD [96]. Exaggerated Ca^2+^ release from RyR has been related to impaired neurophysiology and synaptic signaling events, contributing to memory impairment in AD [33,97,98]. The FAD PS mutations also exaggerate Ca^2+^ release through the RyR, as a result of either increased expression of RyRs or sensitization of the channel activity [99,100]. Furthermore, RyR-mediated Ca^2+^ release upregulates secretases, increasing APP cleavage, Aβ fragments, and plaque deposition, and its blockage leads to Aβ reduction and improved memory impairment [101]. RyRs can also themselves be activated by Ca^2+^, which amplify IP_3_R activity via Ca^2+^-induced Ca^2+^ release mechanisms [102]. Additionally, Aβ aggregates themselves trigger ER Ca^2+^ release through IP_3_Rs and RyRs [103,104]. Recently, stabilization of RyR_2_ macromolecular complex by S107 (Rycal)—a benzothiazepine that prevents the dissociation calstabin2 from the RyR_2_ complex—showed therapeutic potential in vitro and in mouse models of AD in vivo. Application or administration of S107 reversed ER Ca^2+^ leak, reduced APP cleavage and Aβ production, and restored synaptic plasticity and cognitive deficits [105,106].

It has been found that, in cells lacking PS1, PS2, or PS1/2, or cells expressing either PS2 or FAD-PS2, SERCA activity is diminished, resulting in increased cytosolic Ca^2+^ [107]. Conversely, other studies have shown that mutations in PS influence SERCA by accelerating Ca^2+^ sequestration via ATPase [108], leading to an overfilled ER. In any case, these data suggest that normal PSs are required for normal SERCA functioning and suggest that PSs are a candidate target for development of therapeutics, independent of their role in APP processing.

Increased Ca^2+^ release from intracellular ER Ca^2+^ stores might exacerbate disease-mediated pathology. Accordingly, dantrolene, a negative allosteric modulator of RyR—and its central nervous system (CNS)-penetrant version Ryanodex—has been shown to reduce amyloid pathology, normalize ER Ca^2+^ homeostasis, restore synaptic structure and density, normalize synaptic plasticity, and improve behavioral performance in mouse models of AD [101,109,110]. This builds on RyR as a therapeutic target for AD, and further emphasizes the role of dysregulated ER Ca^2+^ as a key component in the AD pathogenesis.

ER Ca^2+^ depletion triggers a sustained extracellular Ca^2+^ influx to the cytosol through the store-operated Ca^2+^ entry (SOCE) pathway by activating STIM (stromal-interacting molecule) protein—which senses low Ca^2+^ concentration upon depletion of the ER stores—and plasma membrane channels Orai and TRPC (transient receptor potential canonical) [111]. Two forms of STIM are expressed in the brain (STIM1, predominantly in the cerebellum, and STIM2 in the hippocampus and cortex) [112]. SOCE refills the ER, keeping it ready for the next ER Ca^2+^ signal [113]. Disrupted SOCE has been observed in AD. SOCE is decreased by diverse FAD PS mutations [114,115] and in the presence of soluble Aβ [116]. It has also been proposed that SOCE deficits may be due to the decreased expression of STIM1 and/or STIM2 in FAD-linked PS1 mutations [117]. Related to this, overexpression of the dominant negative PS1 variant potentiates SOCE [118]. It has also been proposed that SOCE deficits might result from overfilled ER Ca^2+^ stores [114]. These findings, however, are inconsistent, as other groups have observed no differences or decreased ER Ca^2+^ concentration in mutant PS expressing cells [11,93,107]. Recently, it has been proposed that neuronal SOCE is required for maintaining the morphology of mushroom spines, modulating Aβ production and promoting memory functions [117,119]. Ca^2+^ entry via SOCE activates Ca^2+^/CaM-dependent kinase II (CamKII), which is upstream of gene transcription for maintenance of mature spines. Attenuated SOCE-mediated Ca^2+^ influx might reduce CaMKII activity while inducing destabilization of mushroom spines. This can reduce LTP-mediated memory formation [117]. Attenuated SOCE may also lead to inadequate ER refill, which might induce neuronal cell death via apoptosis [120,121]. STIM2 overexpression in AD models restores spine morphology, implicating SOCE in AD [122] and suggesting that targeting SOCE in AD may avoid or restore dendritic spine loss. Additionally, it was recently found that expression of TRPC1, a subfamily of TRPCs, is decreased in AD cells and mouse models. While deletion of TRPC1 did not impair cognitive function or lead to cell death in physiological conditions, it did exacerbate memory deficits and increase neuronal apoptosis induced by Aβ. On the contrary, overexpression of TRPC1 inhibited Aβ production and decreased apoptosis [123]. Together, these studies suggest another mechanistic target for therapeutic development within the Ca^2+^ hypothesis of AD.

### 2.3. Targeting Mitochondrial Ca^2+^

Mitochondria are crucial organelles that provide energy to the cell in the form of adenosine triphosphate (ATP) via the process of oxidative phosphorylation. Mitochondria form a dynamic tubular network that extends throughout the cytosol, undergoing fusion and fission, which regulates the morphology and structure of the mitochondrial network [124]. Neurons rely strictly on mitochondria to produce ATP, with mitochondria being recruited in areas like synapses, where high energy is required. Mitochondria also buffer Ca^2+^ and shape its signal [125], which is involved in neurotransmission and maintenance of the membrane potential along the axon. At the synaptic level, mitochondria regulate the Ca^2+^ levels necessary for synaptic functions [126]. Mitochondrial Ca^2+^ uptake activates some dehydrogenases at the electron transport chain (ETC), activating mitochondrial respiration and ATP production [127]. The electrochemical gradient created by the ETC allows mitochondria to take up Ca^2+^. This mitochondrial Ca^2+^ participates in signal transduction and the production of energy. Mitochondria contain two major membranes, the outer mitochondrial membrane (OMM), which contains voltage-dependent anion-selective channel protein (VDAC), permeable to most molecules, and the inner mitochondrial membrane (IMM), which is impermeable to molecules and ions, unless they contain specific channels or transporters.

Ca^2+^ is taken up into the mitochondrial matrix through the mitochondrial Ca^2+^ uniporter (MCU) complex, a highly Ca^2+^-sensitive ion conductance channel [128,129]. The MCU is a macromolecular complex of proteins, which includes the pore and several regulatory subunits. It is ubiquitously expressed among organisms and defines the pore domain of the complex [128]. Two other proteins participate in the Ca^2+^ permeant pore: MCUb, whose expression is restricted to most vertebrates [130,131], and the essential MCU regulator (EMRE) [132]. The response of the MCU to extramitochondrial Ca^2+^ is regulated by the mitochondrial Ca^2+^ uptake (MICU) family of proteins, which are in the intermembrane space. MICU1 and MICU2 act as Ca^2+^ sensors, each with two Ca^2+^-binding EF-hand motifs that confer sensitivity to Ca^2+^ [133]. MICU1 and MICU2 also act as gatekeepers of MCU [134], with MICU1 getting involved when the extramitochondrial Ca^2+^ concentration is high, activating the channel open state. At low concentrations, the main player seems to be MICU2, leading to minimal accumulation of Ca^2+^ within mitochondria [135,136], thus preventing mitochondrial Ca^2+^ overload at resting conditions. MICU3, a paralog of MICU1 and MICU2, is mainly expressed in the CNS [137], and has been proposed to enhance mitochondrial Ca^2+^ uptake in neurons [138]. In regulating the MCU pore, the mitochondrial Ca^2+^ uniporter regulator 1 (MCUR1) also plays a role [139]. It has been suggested as a necessary player in MCU-mediated mitochondrial Ca^2+^ uptake. The small Ca^2+^-binding mitochondrial carrier protein (SCaMC, also known as SLC25A23) [140] seems to also participate in the mitochondrial Ca^2+^ uptake by interacting with MCU and M1CU1.

Mitochondrial Ca^2+^ efflux occurs via the Na^+^/Ca^2+^ exchanger (NCLX) [141] and leucine zipper- and EF hand-containing transmembrane protein 1 (Letm1), located at the IMM [142]. Excessive Ca^2+^ in the mitochondrial matrix induces the activation of the mitochondrial permeability transition pore (mPTP) and allows the release of Ca^2+^ ions and small molecules such as cytochrome c [143]. Mitochondrial Ca^2+^ levels are tightly regulated since excessive levels of Ca^2+^ within mitochondria, i.e., mitochondrial Ca^2+^ overload, result in the impairment of mitochondrial function, suppression of ATP production, increase in reactive oxygen species (ROS) production, and mPTP opening. This can lead to caspase activation and cell death via apoptosis [144].

Mitochondrial function has long been considered one of the intracellular processes compromised at the early stages in AD and likely in other neurodegenerative diseases. Moreover, the “mitochondrial cascade hypothesis” was proposed to explain the onset of SAD [145], which posits that mitochondrial dysfunction is the primary process to trigger the cascade of events that lead to late-onset AD. Even though the validity of this hypothesis has yet to be demonstrated, numerous mitochondrial functions are disrupted in AD [146], including mitochondrial morphology and number [147], oxidative phosphorylation, mitochondrial membrane potential, ROS production [148], mitochondrial DNA (mtDNA) oxidation and mutation [149], mitochondrial–ER contacts [150], and mitochondrial dynamics, including mitochondrial transport along the axon and mitophagy [151]. Additionally, both Aβ and tau have been found in mitochondria. Aβ is imported to mitochondrial matrix via translocase of the outer membrane (TOM) [152], and a fraction of intracellular tau has been found within the inner mitochondrial space [153]. Once in mitochondria, they interact with specific intramitochondrial targets, leading to the dysfunction of the organelle. Furthermore, tau accumulation in mitochondrial synaptosomes has been proposed to correlate with synaptic loss in AD brains [154].

Mitochondrial Ca^2+^ dysregulation is considered a fingerprint of AD. Mitochondrial Ca^2+^ overload can be a result of three different processes: (i) increased mitochondrial Ca^2+^ influx (following Ca^2+^ influx from extracellular space or Ca^2+^ transfer from ER), (ii) decreased mitochondrial Ca^2+^ efflux through NCLX, or (iii) reduced mitochondrial Ca^2+^ buffering. Neurotoxic Aβ can lead to mitochondrial Ca^2+^ overload, as shown in in vitro and in vivo models [155,156,157]. Primary neurons in culture exposed to Aβ oligomers triggered mitochondrial Ca^2+^ overload, leading to mPTP opening, release of cytochrome c, and cell death via mitochondrial-mediated apoptosis [156]. Additionally, studies in mouse neuroblastoma N2a cells co-transfected with the Swedish mutant APP and Δ9 deleted PS1 showed similar mitochondrial impairment, evidenced by the increased mitochondrial apoptotic pathway and caspase-3 activity [158]. Furthermore, Aβ can interact with cyclophilin D—a regulator of mPTP—and promote the release of cytochrome c through the opening of mPTP [159]. This causes neuronal injury and decline of cognitive functions, as shown in a mouse model of AD. Genetic deletion of CypD in Tg AD mice rescues mitochondrial impairment and improves learning and memory [160], suggesting that CypD could represent a potential therapeutic target in AD.

Recently, we showed mitochondrial Ca^2+^ overload in a mouse model of cerebral amyloidosis (APP/PS1). Using in vivo multiphoton imaging and a ratiometric Ca^2+^ reporter, we demonstrated increased levels of mitochondrial Ca^2+^ following Aβ deposition, which preceded neuronal cell death. Moreover, naturally secreted soluble oligomers applied to the healthy brain of Wt mice also increased mitochondrial Ca^2+^ levels, a process that could be prevented by MCU inhibition with the specific channel blocker Ru360 [157]. We also showed, for the first time, that the expression of mitochondrial Ca^2+^ transport-related genes in brain tissue from AD patients was impaired compared to control cases. In particular, genes involved in mitochondrial Ca^2+^ uptake (MCU complex) were downregulated, whereas the only one encoding for Ca^2+^ efflux (NCLX) was upregulated, suggesting a compensatory response to prevent mitochondrial Ca^2+^ overload [157]. However, others reported that different techniques used for evaluating expression showed conflicting results [161]. Another mechanism proposed for mitochondrial Ca^2+^ overload in AD is impairment of mitochondrial Ca^2+^ efflux. Loss of NCLX expression and functionality has also been suggested in AD, whereas genetic rescue of NCLX expression in neurons restored cognitive decline and cellular impairment in transgenic mouse models of AD [161]. Additionally, the more general cytosolic Ca^2+^ overload observed in vivo (as previously cited) may contribute to the observed mitochondrial Ca^2+^ overload. These observations suggest that restoring mitochondrial Ca^2+^ levels in AD could be a promising new therapeutic target against AD.

It has been previously proposed that nonsteroidal anti-inflammatory drugs (NSAIDs) may help in preventing the cognitive decline associated with aging [162]. Unfortunately, results from several clinical trials have given rather pessimistic results [163,164,165], partly due to inadequate CNS drug penetration of existing NSAIDs, suboptimal doses, unknown molecular targets (and, therefore, unknown pharmacodynamics), and toxicities. Nevertheless, in vitro studies have shown that NSAIDs such as salicylate and the enantiomer (*R*)-Flurbiprofen lacking anti-inflammatory activity, at low concentrations, are able to depolarize mitochondria and inhibit the driving force for mitochondrial Ca^2+^ uptake [166,167]. They act as mild mitochondrial uncouplers without altering cytosolic Ca^2+^ levels. This mild mitochondrial depolarization was able to prevent NMDA- and Aβ-induced mitochondrial Ca^2+^ uptake and cell death [155,156,168]. These results point to mitochondrial Ca^2+^ as a key player in Aβ-driven neurotoxicity and suggest a new mechanism of neuroprotection by NSAIDs independent of their anti-inflammatory activity. Another compound, TG-2112x, has been recently suggested as neuroprotective and proposed as a new therapeutic opportunity. Tg-2112x partially inhibits mitochondrial Ca^2+^ uptake without affecting the mitochondrial membrane potential or mitochondrial bioenergetics, protecting neurons against glutamate excitotoxicity [169].

Abnormal tau hyperphosphorylation also influences mitochondrial transport along the neuronal axon, which leads to a reduction in and impairment of mitochondria at the presynaptic terminal with detrimental consequences and eventual cell death [170,171]. In vitro and in vivo studies have shown that tau dysregulates Ca^2+^ homeostasis in mitochondria. Mitochondrial Ca^2+^ buffering and homeostasis are disrupted in cells overexpressing tau and those exposed to extracellular tau aggregates [61,172]. Additionally, basal mitochondrial Ca^2+^ levels have been shown to be elevated in patient-derived human induced pluripotent stem cell (iPSC) neurons expressing a tau mutation, likely due to the inhibition of NCLX by tau [173]. Elevation in mitochondrial Ca^2+^ levels by tau also increased the vulnerability to Ca^2+^-induced cell death [173]. Phosphorylated tau has also been found to interact with VDAC in AD brains, leading to mitochondrial dysfunction [174].

Mitochondria and ER membranes are juxtaposed and establish contact points known as mitochondrial-associated membranes (MAMs). They are dynamic lipid rafts enriched in cholesterol and sphingomyelin, as well as in proteins associated with Ca^2+^ dynamics [175,176]. MAMs allow for communication between ER and mitochondria, including metabolic pathways and Ca^2+^ transfer from ER to mitochondria [177]. Increased contacts between ER and mitochondria have been found in human fibroblast cells derived from FAD patients, human brain tissue, and AD mouse models [178,179]. An increased association between the ER and mitochondria has also been observed in a Tg mouse model of tauopathy [180]. Increased contact promotes mitochondrial bioenergetics, but excessive Ca^2+^ transfer can contribute to mitochondrial Ca^2+^ overload and suppression of normal mitochondrial functions, and Aβ oligomers have been found to induce massive Ca^2+^ transfer from ER to mitochondria [116,181,182,183].

Mitochondria-targeted protective compounds that prevent or minimize mitochondrial dysfunction could represent potential therapeutic strategies in the prevention or treatment of AD. However, several compounds targeting mitochondrial function have been tested in AD without a favorable outcome [184]. Nevertheless, the idea of AD as a multifactorial disease is widespread, and mitochondria as a therapeutic target combined with other medications is emerging as a valid therapy for AD. The list of pharmacologic approaches that directly target mitochondria includes antioxidants (such as vitamin E and C, coenzyme Q10, mitoQ, and melatonin) and phenylpropanoids (such as resveratrol, quercetin, or curcumin) [185]. Antioxidants are generally used to decrease oxidative stress and slow the progression of symptoms that generally accompany AD. Antioxidants such as coenzyme Q10 and mitoquinone mesylate (MitoQ) are antioxidants that directly target mitochondria [186]. Currently, there is a small clinical trial testing MitoQ on cerebrovascular blood flow in AD [187]. The Szeto-Schiller (SS) tetrapeptides, an alternative type of antioxidants that target mitochondria, are small molecules that can reach the mitochondrial matrix and act as antioxidants [188]. Specifically, SS31 (also known as elamipretide) selectively binds to cardiolipin and promotes electron transport while optimizing mitochondrial ATP synthesis [189]. In addition, SS31 inhibits mitochondria swelling and oxidative cell death. In mouse models of cerebral amyloidosis, it was shown that SS31 reduces Aβ production and mitochondrial dysfunction, and enhances mitochondrial biogenesis and synaptic activity [190]. Recently, SS31 combined with the mitochondrial division inhibitor 1 (Mdivi1) was tested in vitro with a positive outcome, suggesting this combination as a possible type of mitochondria-targeted antioxidant in AD [191]. Ongoing clinical trials regarding mitochondria in AD are reviewed in [187,192] and at www.clinicaltrials.gov.

### 2.4. Targeting Lysosomal Ca^2+^

Lysosomes are acidic organelles that participate in the endolysosomal system. They are important for autophagy and intracellular Ca^2+^ storage (with comparable Ca^2+^ levels to those of the ER) [193]. The Ca^2+^ transport in and out of the lysosomal lumen provides signals that modulate the fusion of autophagosomes and lysosomes. In order to maintain lysosomal Ca^2+^ homeostasis, lysosomes contain P/Q type VGCCs expressed in the lysosomal membrane that provide Ca^2+^ to the cytosol. Dysregulation in lysosomal Ca^2+^ release via VGCCs leads to defective autophagic fusion and flux [194]. The vacuolar-type H^+^-ATPase (V-ATPase) and Ca^2+^/H^+^ exchanger are in charge of lysosomal Ca^2+^ refilling [195]. It has been suggested that this refilling is largely dependent on ER Ca^2+^ [196]. V-ATPase activity predominantly maintains lysosomal pH; however, other ion channels localized to the lysosomal membrane participate in pH regulation during lysosomal proteolysis, including the chloride channel CLC7 [197] and the Ca^2+^ channel TRPML1 (mucolipin) [198,199]. Additionally, Ca^2+^ microdomains generated at the mouth of these channels have been suggested to take part in the regulation of autophagy [200].

Lysosomal Ca^2+^ efflux has been linked to changes in lysosomal pH. Recent reports suggested that decreased lysosomal Ca^2+^ in AD-linked mutations or PS1 knockout (KO) cells is a consequence of elevated lysosomal pH [201]. Raising lysosomal pH leads to autophagy defects and lysosomal Ca^2+^ efflux. PS1 mutant cells exhibit these defects. PS1 KO cells show deficiencies in lysosomal V-ATPase content and function, defective autophagy, and abnormal Ca^2+^ efflux. [201]. Reversal of lysosomal pH abnormalities in PS1 KO cells, but not Ca^2+^ efflux deficits, was sufficient to rescue these same deficits [201]. These data suggest that lysosomal Ca^2+^ defects are secondary to lysosomal pH elevation, and that lysosomal Ca^2+^ dyshomeostasis contributes significantly to the overall Ca^2+^ dysregulation observed in PS1-deficient cells. However, other studies do not support these arguments, citing that, although the autophagosome and lysosome accumulation was apparent in PS1 or PS2 cells, defective lysosome acidification was not found [202]. In addition, defects in lysosome acidification or Ca^2+^ homeostasis have not been observed in FAD-PS2 models [203]. On the contrary, other studies have shown both reduced cytosolic Ca^2+^ signal and lower ER content in FAD-PS2 models. In particular, it was proposed that FAD-PS2 decreases ER and *cis*–medial Golgi Ca^2+^ levels by reducing SERCA activity, which could lead to defective autophagosome–lysosome fusion [203]. Further studies are necessary to confirm these observations and demonstrate whether or not lysosomal Ca^2+^ or pH could be potential therapeutic targets for AD.

Autophagy is a lysosomal degradative pathway responsible for the recycling of different cellular constituents. Especially important under conditions of metabolic stress, this pathway aids in the cellular turnover of damaged or obsolete organelles in order to eliminate misfolded and aggregated proteins left behind by the ubiquitin-proteasome system [204]. Materials are engulfed within double-membrane vesicles (autophagosomes) and targeted to lysosomes for degradation of molecular components. Disruption of autophagy results in accumulation of autophagic vacuoles within swollen dystrophic neurites of affected neurons [205]. Lysosomal Ca^2+^ has been proposed to trigger transcriptional activation of autophagic proteins [200]. Impairment of the autophagy–lysosomal pathway has been described as a hallmark of AD related to lysosomal Ca^2+^ dyshomeostasis. This dysregulation impacts clearance of Aβ and hyperphosphorylated tau, and contributes to their accumulation in the brain [206,207].

## 3. Astrocytic Ca^2+^ as a Therapeutic Target in AD

Astrocytes, the most abundant cells in the brain, are key regulators of molecular homeostasis in the nervous system. They provide trophic and metabolic support to neurons, sense and modulate neuronal network excitability, and participate in neurovascular coupling and maintenance of the blood–brain barrier [208,209,210]. Astrocytes do not generate action potentials, but exhibit Ca^2+^ transients followed by a release of gliotrasmitters—such as ATP, glutamate, or gamma-aminobutyric acid (GABA)—in response to neurotransmitters [211]. It has been proposed that the Ca^2+^ global signals—propagating waves—rely on Ca^2+^ release from the ER (mostly mediated by IP_3_R). Local Ca^2+^ microdomains, on the other hand, result from Ca^2+^ influx via ionotropic receptors, TRPs, SOCE, mitochondrial Ca^2+^ activity, or reversed Na^+^/Ca^2+^ exchangers [212].

In AD, astrocytes become activated. Reactive astrogliosis is characterized by the biochemical, functional, and morphological reshaping of astrocytes aimed at neuroprotection [213]. Reactive astrocytes upregulate activation markers such as glial fibrillary acidic protein and vimentin. Using postmortem human tissue, it has been shown that reactive astrocytes associated with plaques express higher levels of the glutamate metabotropic receptor mGluR5, which induces Ca^2+^ release form intracellular stores [214]. In vitro, exposure of astrocytes to Aβ increases basal intracellular Ca^2+^ levels as a result of extracellular Ca^2+^ entry, release from mGluR5 and IP_3_R, and induced Ca^2+^ oscillations or transients [214,215]. Pharmacological inhibition of ER Ca^2+^ release blocks the Aβ-induced astrogliosis both in cultured astrocytes and in organotypic slices [216]. As observed in co-cultures of neurons and astrocytes, the Aβ-induced astrocytic Ca^2+^ transients are followed by neuronal death, suggesting that aberrant astrocytic Ca^2+^ signal results in neurotoxicity [217]. These results, however, are not universal, and other groups have not replicated these observations [218]. It has also been suggested that *APOE4* dysregulates Ca^2+^ excitability in astrocytes by modifying membrane lipid composition. This phenomenon was observed in hippocampal slices from *APOE3* and *APOE4* mice, specifically in male mice [219], suggesting that the *APOE* genotype modulates Ca^2+^ fluxes in astrocytes in a lipid and sex-dependent manner. As demonstrated in primary cortical co-cultures of neurons and astrocytes, exposure to insoluble aggregates of tau failed to induce a Ca^2+^ response in astrocytes [36]. Unfortunately, little else is known about the effects of tau on astrocytic Ca^2+^, and further research is clearly warranted.

In the intact brain in vivo under physiological conditions, astrocytes show sporadic Ca^2+^ transients as a hallmark of astrocytic activity [220]. As demonstrated in cortical astrocytes of amyloid-depositing mice (APP/PS1), under pathological conditions, the frequency of spontaneous Ca^2+^ waves increases [18]. These same astrocytes exhibit higher resting Ca^2+^ levels. While overall astrocytic hyperactivity was noticed throughout the cortical tissue and not just in the vicinity of amyloid plaques, the astrocytes initiating the intracellular Ca^2+^ waves were located in plaque vicinity [18]. Further studies are needed to determine whether this was an effect of soluble Aβ oligomers or Aβ fibrils. Ca^2+^ hyperactivity in astrocytes has been associated with abnormal purinergic signaling, suggesting that reactive astrocytes release excessive amounts of ATP. This in turn activates P2Y purinoceptors mediating abnormal cytosolic Ca^2+^ signaling [17]. It has also been suggested that alterations in extracellular Ca^2+^ levels can be involved in astrocytic hyperactivity. During increased neuronal activity, extracellular Ca^2+^ decreases following ionotropic glutamate receptor and VGCC activation. Astrocytes sense the extracellular Ca^2+^ decrease and release ATP in response [221]. Increases in extracellular ATP trigger astrocytic Ca^2+^ transients and could contribute to AD-associated astrocytic hyperactivity.

## 4. Microglial Ca^2+^ as a Therapeutic Target in AD

Microglia are the major immune cells in the brain. They sense and react to alterations in brain homeostasis. They are also involved in synaptic pruning, which occurs during the first weeks of postnatal development and is critical for the maturation of neuronal networks [222]. Microglial activation is characterized by morphological alterations and production of pro- and anti-inflammatory mediators [223]. Intracellular Ca^2+^ regulates microglial activation from its homeostatic resting state to a neurotoxic-activated state. Some microglial functions, including the production and release of proinflammatory factors, such as nitric oxide (NO) and certain cytokines, are Ca^2+^-dependent processes [224]. In turn, proinflammatory cytokines, tumor necrosis factor α (TNFα), interleukin 1β, and interferon γ, all increase intracellular Ca^2+^ levels in microglia [225,226,227], while anti-inflammatory cytokines decrease them [228].

AD has long been linked to microglial activation. Microglia surround amyloid plaques [229] after they get recruited within the first days after plaque formation [230]. Once activated, microglia internalize and break down Aβ. Microglia activation is an early process in AD, and it has been shown to be correlated with cognitive deficits [231]. Released proinflammatory cytokines (such as IL-1β and TNF-α) by microglia might stimulate the release of proinflammatory substances by astrocytes, amplifying the inflammatory signal and its neurotoxicity [232,233]. Therefore, neurons and astrocytes in the vicinity of these plaques are likely subjected to high levels of proinflammatory mediators released by activated microglia. These mediators can cause alterations in the Ca^2+^ homeostasis of these cells [234]. Additionally, microglial cultures exposed to Aβ increase their immune response (i.e., cytokine production) and intracellular Ca^2+^, a process that can be blocked by the dihydropyridine nifedipine and the non-dihydropyridine L-type VGCC antagonist verapamil or diltiazem [235].

Observations from in vitro data have shown that intracellular Ca^2+^ homeostasis is impaired in activated microglia. Microglia isolated from AD brain tissue have elevated cytosolic Ca^2+^ levels compared to controls and exhibit reduced responsiveness to stimuli in vitro [236]. Additionally, mouse microglia activated by lipopolysaccharide (LPS) display increased basal Ca^2+^ levels and a reduced agonist-induced Ca^2+^ signal [224]. Ramified activated microglia display large intracellular Ca^2+^ transients in response to the damage of individual cells in their vicinity. The use of in vivo Ca^2+^ imaging and multiphoton microscopy has allowed the study of microglial Ca^2+^ dynamics in the intact living brain. Microglia display rare Ca^2+^ transients in their resting state, but respond with larger Ca^2+^ transients when activated [237]. Microglial Ca^2+^ transients are attributed to Ca^2+^ release from intracellular stores and are prevented by the activation of ATP receptors [237]. Blocking SOCE—via knocking down or knocking out STIM1/2 and Orai—reduces immune functioning, including phagocytosis, migration, and cytokine production in primary isolated murine microglia [238,239]. On the other hand, blocking RyR prevents LPS-induced neurotoxicity mediated by microglia [240]. Although they have not been studied in depth, it has been suggested that microglia display Ca^2+^ microdomains. Some observations suggest that global Ca^2+^ elevations in microglia trigger phagocytosis and migration, whereas local Ca^2+^ increases in their processes regulate acute chemotactic migration [241]. Taken together, elevated Ca^2+^ levels seem to be a hallmark of activated microglia and their regulation, a potential therapeutic target for AD therapy.

Microglia express P2X receptors, a subfamily of purinergic ionotropic receptors, located in the plasma membrane which are permeable to Ca^2+^, Na^+^, and K^+^ [242]. Overactivation of P2X receptors may lead to cell death via membrane depolarization, mitochondrial stress, and ROS production [243]. As measured in postmortem brain tissue from AD patients, P2X_7_ expression is upregulated in microglia in AD [244]. This same effect was shown in plaque-associated microglia in an AD mouse model and after intrahippocampal injection of Aβ_42_ [245]. It is believed that these high levels of P2X_7_ contribute to the enhanced inflammatory responses observed in AD [244,246,247]. Inhibition of P2X_7_ receptors has been shown to be neuroprotective as it reduces the dendritic spine loss induced by Aβ [248], as well as Aβ production in general [249]. P2X_7_ KO mice express reduced plaque size and improved behavioral scores [250], suggesting P2X_7_ as a potential therapeutic target in AD. Additional in vitro studies have shown that P2X_7_ activation leads to microglial NOD-, LRR- and pyrin domain-containing protein 3 (NLRP3) inflammasome activation, which requires Ca^2+^ mobilization from intracellular stores [251,252]. Aβ also triggers this NLRP3 activation [253], and NLRP3 is highly activated in microglial cells surrounding amyloid plaques [253]. NLRP3 KO mice show reduced amyloid burden in the brain and have reduced memory impairment [254]. This confirms that P2X_7_ and NLRP3 could be candidate targets for AD therapeutics. 

The triggering receptor expressed on myeloid cells 2 (*TREM2*) gene has been recently identified as a risk gene for AD [255]. Its low-frequency variants increase the risk of developing AD similar to the APOE4 allele. TREM2 is a transmembrane protein receptor expressed on microglia. It stimulates phagocytosis and suppresses inflammation [255]. TREM2 overexpression in a mouse model of AD (APP/PS1) decreased AD-related pathology and improved cognitive functions [256], suggesting that modeling microglial functions could be a protective target in AD. Immunotherapy using antibodies to stimulate TREM2 signaling in order to improve AD pathology is currently being developed by different groups. Stimulation with anti-TREM2 antibodies in vitro produced Ca^2+^ influx and extracellular signal-regulated kinase (ERK) signaling activation in human dendritic cells [257]. When to stimulate TREM2 to treat AD, however, is not clear, and it must be kept in mind that the use of these antibodies could alter the binding of other TREM2 ligands. Further studies will be needed to fully understand TREM2 function and its role in AD therapy. 

## 5. Conclusions and Future Directions

AD is a multifactorial complex disease that leads to progressive dementia. Its nature brings upon the equally complex task of developing a treatment strategy. Current medications for Alzheimer’s disease only treat the symptoms and cannot stop the damage that AD pathology causes to brain cells. Therefore, an urgent need exists for new target discovery that directly targets AD pathology and alters the course of its progression. On the basis of a wide range of studies, evidence suggests that treatment should be initiated in AD’s earliest stages, before the start of deposition of pathology and occurrence of irreversible mental decline. Ca^2+^ dyshomeostasis is an early event in the AD timeline. Ca^2+^ dysregulation in AD comes as a result of hyperactivity of Ca^2+^ channels in the plasma membrane and intracellular compartments. It does not seem to be restricted to neurons, but rather is a global phenomenon that affects many cell types in the brain (Figure 2).

Intracellular Ca^2+^ homeostasis is mediated by several organelles, such as the ER, mitochondria, and lysosomes, which contribute to cell stress regulation. Increased Ca^2+^ concentrations in these compartments disrupt normal homeostasis, eventually leading to accumulated pathogenic proteins, which in turn further impair Ca^2+^ homeostasis, leading to severe alterations in neuronal circuitry. With these underlying data, it is clear that isolating potential therapeutic strategies aimed at normalizing Ca^2+^ levels is important. Current FDA-approved AD treatments target plasma Ca^2+^ channels, but more specific approaches are needed to target other prevalent and disrupted intracellular Ca^2+^ signaling pathways, such as those of the ER or mitochondria. As our knowledge in Ca^2+^ dysregulation in AD grows, it seems more obvious that targeting these other sources of Ca^2+^ dysregulation could be an effective therapeutic strategy.

A better understanding of the onset and progression of neurodegenerative diseases will facilitate rapid diagnosis and target selection, allowing for early treatment. A truly effective method for preventing or treating Alzheimer’s disease will likely involve a combination approach for targets, such as Aβ plaque clearance or soluble tau removal. Additionally, reversal of cellular processes that are disrupted by Aβ or tau accumulation (including Ca^2+^ dyshomeostasis), early diagnosis, and/or lifestyle changes would also be necessary for successful therapeutic intervention. Gene therapy is an emerging therapeutic strategy for the treatment of neurodegenerative disorders, including AD, particularly when traditional therapies are not responsive to well-validated genetic targets. Gene therapy has already shown efficacy in preclinical studies, utilizing different routes for gene delivery [258,259]. Recently, different groups proposed gene therapy as a strategy in the battle against AD, as it is designed to focus on one specific target in affected brain regions. Several gene therapy strategies for AD have already been tested. These include acting directly on APP metabolism, neuroprotection, targeting inflammatory pathways, or modulating genes related to lipid metabolism [260]. Unfortunately, they have not provided an encouraging outcome so far, as they sometimes show unexpected or undesirable side effects. One of the ongoing clinical trials is designed to evaluate gene therapy use in AD patients (already clinically diagnosed) that are APOE4 homozygotes (www.clinicaltrials.gov). The study aims to evaluate whether intracisternal administration of APOE2 to APOE4 homozygotes AD patients will lead to conversion of the APOE protein isoforms from APOE4 homozygotes to APOE2–APOE4, which has given positive results in mice and monkeys previously [261,262]. If this therapy slows the illness in people with advanced AD, this could also function as a method for disease prevention, reducing the risk of disease development in healthy people. A combination of these targets and therapies should reduce stress levels and cell death in AD, offering pathological and potentially symptomatic relief.

## Figures and Tables

**Figure 1 cells-09-02513-f001:**
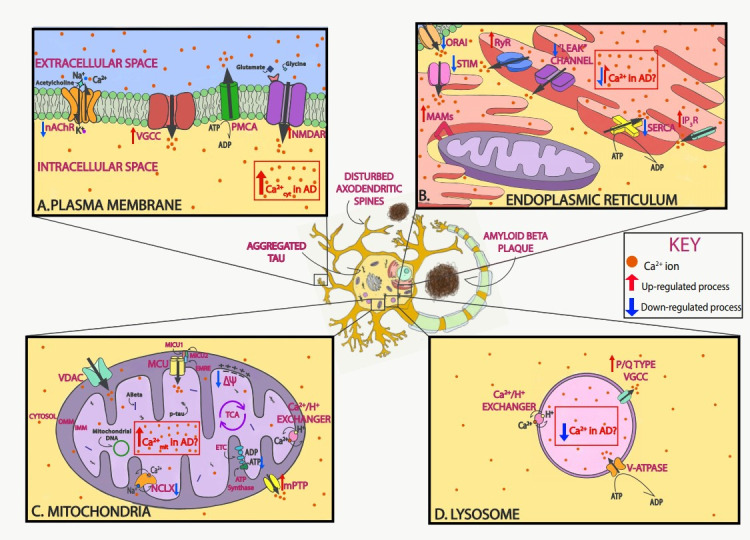
Neuronal Ca^2+^ as a therapeutic target in Alzheimer’s disease (AD). Schematic of Ca^2+^ dysregulation in neurons in AD that could be used as potential targets. In AD, Ca^2+^ dysregulation is present in many of the different compartments within neurons. In the plasma membrane, voltage-gated Ca^2+^ channels (VGCCs) and receptor operated Ca^2+^ channels, including *N*-methyl-d-aspartate receptors (NMDARs) and nicotinic acetylcholine receptors (nAChRs), allow for the influx of Ca^2+^ ions into the neuron after depolarization or ligand binding, respectively. Both Aβ and tau overactivate these channels and increase their function (**A**). In the endoplasmic reticulum (ER), Ca^2+^ is released via ryanodine receptors (RyRs) and inositol 1,4,5-trisphosphate receptors (IP_3_Rs) to the cytosol after stimulation. Ca^2+^ is then extruded by the sarco-endoplasmic reticulum ATPase (SERCA) pump, which actively consumes ATP while bringing Ca^2+^ into the lumen. AD-associated presenilin (PS) mutations impair IP_3_R and RyR signaling, increasing Ca^2+^ release into the cytosol, and diminish SERCA activity, increasing cytosolic Ca^2+^ concentration. Following ER Ca^2+^ depletion, the stromal-interacting molecule (STIM) interacts with the Orai channel in the plasma membrane to activate the store-operated Ca^2+^ entry (SOCE) pathway. SOCE is decreased by diverse familial AD (FAD) PS mutations and by soluble Aβ. Lastly, in order to facilitate the communication between mitochondria and ER, contact points known as mitochondrial-associated membranes (MAMs) are established. Increased association between the ER and mitochondria and enhanced Ca^2+^ transfer have been observed in AD (**B**). In the mitochondria, the voltage-dependent anion-selective channel protein (VDAC) lets Ca^2+^ across the outer mitochondrial membrane (OMM), and the mitochondrial Ca^2+^ uniporter (MCU) complex allows the influx of Ca^2+^ across the inner mitochondrial membrane (IMM). Ca^2+^ efflux is partially managed by the Na^+^/Ca^2+^ exchanger (NCLX). Both Aβ and tau (phospho-tau, p-tau) have been found in mitochondria. Elevated mitochondrial Ca^2+^ levels and decreased NCLX activity have been observed in AD (**C**). In the lysosome, the P/Q type VGCCs in their membrane regulate Ca^2+^ efflux into the cytosol, while the V-ATPase and Ca^2+^/H^+^ exchanger are in charge of lysosomal Ca^2+^ refilling (**D**). Additionally, Aβ and tau accumulate extracellularly and intracellularly, respectively, and lead to loss of dendritic spine density and synaptic function.

**Figure 2 cells-09-02513-f002:**
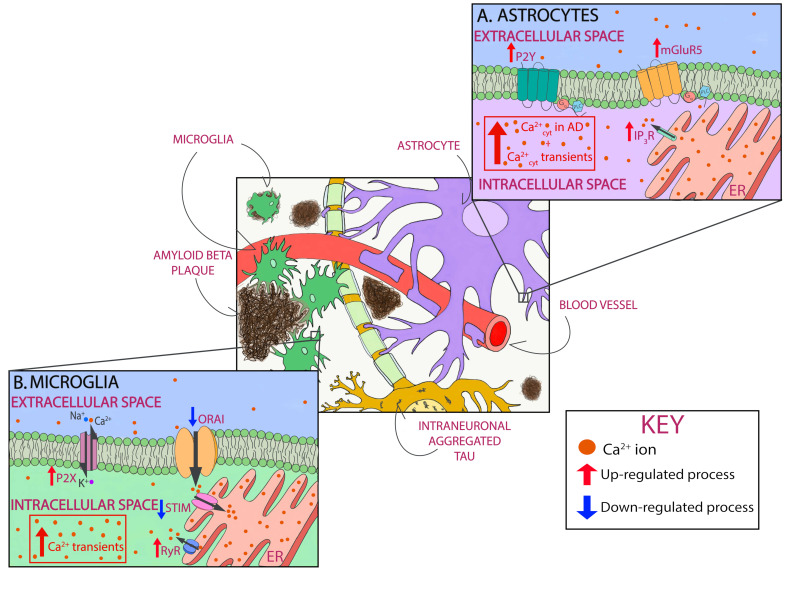
Astrocytic and microglial Ca^2+^ as a therapeutic target in AD. Schematic of glial Ca^2+^ cells dysregulation in the presence of AD pathology. In astrocytes, P2Y purinoceptors and glutamate metabotropic receptors mGluR5, when activated, cause Ca^2+^ increase by releasing Ca^2+^ from intracellular stores. As shown in red, all three receptors are upregulated in AD. In addition, cytosolic Ca^2+^ levels are increased in astrocytes, and they exhibit Ca^2+^ transients (**A**). In microglia, P2X receptors are upregulated in AD, thus leading to Ca^2+^ dysregulation. SOCE, with involves STIM and Orai, is also responsible for Ca^2+^ influx, specifically into the lumen of the endoplasmic reticulum (ER). This pathway is downregulated in AD (shown in blue). RyRs mediate Ca^2+^ efflux from the ER, a process that is upregulated in AD. Microglia also show Ca^2+^ dysregulation by showing cytosolic Ca^2+^ transients (**B**).

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
