# Peer review of "Therapeutic Strategies to Target Calcium Dysregulation in Alzheimer’s Disease"

_cells, 2020, doi:10.3390/cells9112513_

Round 1

Reviewer 1 Report

The review describes possible therapeutic strategies that may be used to target calcium dysregulation in Alzheimer’s disease. This a very well-written review, which is timely. There are a few minor points which require the authors' attention.

1) More background information on presenilin is needed.

2) TRPC stands for Transient Receptor Potential Canonical (see Cells 2020, 9(9), 1983)

3) The TRPC1 protein was reported to be beneficial in AD (Journal of Alzheimer's Disease, vol. 63, no. 2, pp. 761-772, 2018). This should be discussed in the review.

4) Please revise the following statement “Two membranes compose mitochondria,…”

5) Line 448: “used” is missing.

6) Please define “NCLX “

7) Line 557: Please revise the following sentence to improve clarity: “However, there are studies do not support these arguments, citing that 557 although the autophagosome and lysosome accumulation was apparent in PS1 or 558 PS2 cells, defective lysosome acidification was not found [171].”

8) Figure 1B: It may be good to show that Abeta may accumulate in mitochondria.

9) Figure 1C: Is not ATP produced in the matrix of mitochondria? The figure shows that ATP is produced in the intermembrane space… ETC is responsible for establishing low pH in the intermembrane space but does not synthesize ATP. ACC should be shown.

10) Figure 1D Lysosome: This reviewer believes that it should be “H+” rather “Na+” on the diagram of Ca2+/H+ exchanger. The figure shows that lysosomal Ca2+ concentration is increased in AD. This statement requires a reference.  

Author Response

1) More background information on presenilin is needed.

Response: We have extended the background information about PSs, as follows (lines 286-295): “PSs are transmembrane proteins found in the ER membranes and form the catalytic core of the γ-secretase complex that processes APP and other of type 1 transmembrane proteins, like Notch. PSs are essential for learning and memory, and neuronal survival during aging in the murine cerebral cortex. Mutations in PSs have been shown to affect APP processing, leading to increased production of the more hydrophobic neurotoxic form Aβ42, or increasing the Aβ42/40 production ratio. It has also been proposed that PSEN mutations cause a loss of presenilin function in the brain, triggering neurodegeneration and dementia in FAD”.

2) TRPC stands for Transient Receptor Potential Canonical (see Cells 20209(9), 1983)

Response: Reviewer is right. TRPC has been corrected and now it stands for transient receptor potential canonical.  

3) The TRPC1 protein was reported to be beneficial in AD (Journal of Alzheimer's Disease, vol. 63, no. 2, pp. 761-772, 2018). This should be discussed in the review.

Response: We appreciate the comment. The link between TRPC1 and AD has now been added to the review and states as follows (lines 361-366): “Additionally, it was recently found that expression of TRPC1, a subfamily of TRPCs, is decreased in AD cells and mouse models. While deletion of TRPC1 did not impair cognitive function or lead to cell death in physiological conditions, it did exacerbate memory deficits and increase neuronal apoptosis induced by Aβ. On the contrary, overexpression of TRPC1 inhibited Aβ production and decreased apoptosis”.

4) Please revise the following statement “Two membranes compose mitochondria,…”

Response: Statement has been replaced by “Mitochondria contain two major membranes” to avoid confusion.

5) Line 448: “used” is missing.

Response: “Used” has been added to the sentence.

6) Please define “NCLX “

Response: NCLX was defined in line 389 as follows: “Mitochondrial Ca2+ efflux occurs via the Na+/Ca2+ exchanger (NCLX) [114] and Leucine zipper- and EF hand- containing transmembrane protein 1 (Letm1), located at the IMM [115]”.

7) Line 557: Please revise the following sentence to improve clarity: “However, there are studies do not support these arguments, citing that 557 although the autophagosome and lysosome accumulation was apparent in PS1 or 558 PS2 cells, defective lysosome acidification was not found [171].”

Response: Sentence has been revised and now states as follows: “However, other studies do not support these arguments, citing that although the autophagosome and lysosome accumulation was apparent in PS1 or PS2 cells…”

8) Figure 1B: It may be good to show that Abeta may accumulate in mitochondria.

Response: Figure 1B has been modified to show Aβ accumulation within mitochondria.

9) Figure 1C: Is not ATP produced in the matrix of mitochondria? The figure shows that ATP is produced in the intermembrane space… ETC is responsible for establishing low pH in the intermembrane space but does not synthesize ATP. ACC should be shown.

Response: Figure 1C has been modified to show that ATP is produced by the ATP Synthase (complex V) in the mitochondrial matrix. Additionally, TCA has been added to mitochondria.

10) Figure 1D Lysosome: This reviewer believes that it should be “H+” rather “Na+” on the diagram of Ca2+/H+ exchanger. The figure shows that lysosomal Ca2+ concentration is increased in AD. This statement requires a reference.  

Response: Figure 1D has been corrected to show H+. Additionally, increased Ca2+ concentration in lysosomes has been also corrected in the figure and reference added in the text (Lee, J.H., et al. (2015) Cell reports 12, 1430-1444).

Reviewer 2 Report

In this manuscript, Calvo-Rodriguez et al review the role of neuronal and glial calcium homeostasis in neurodegeneration and Alzheimer´s disease (AD). They discuss the different proteins involved in AD-related calcium signaling/homeostasis and interventions in their (dysregulated) activity. This review is well written, clearly assigns relevant targets, and critically evaluates the therapeutic strategies in AD.  

Minor comments:

Line 135: CALHM1 is not related to the “classic” voltage-gated calcium channels (L- to T-type channels). Therefore, the term “VGCC” is misleading. I would prefer “calcium channel”.

Line 332: “TRPC” means “classic or canonical TRP channels” to discriminate TRPC channels from the subfamilies TRPV, TRPM etc. Here, the abbreviation “TRP” is suitable.

Lines 677-679: Michaelis et al (Glia, 2015, 63: 652) showed that knockout of STIM1/2 and Orai1 abolishes microglial SOCE and reduces phagocytosis and migration.

Figures: The font size within the figures could be more consistent. For instance, a very big font is used for “PLASMA MEMBRANE” and “ASTROCYTES”, but “Na+” and “PLC” are tiny.

Author Response

Line 135: CALHM1 is not related to the “classic” voltage-gated calcium channels (L- to T-type channels). Therefore, the term “VGCC” is misleading. I would prefer “calcium channel”.

Response: Definition of CALHM1 has been corrected and now states “Ca2+ channel”.

Line 332: “TRPC” means “classic or canonical TRP channels” to discriminate TRPC channels from the subfamilies TRPV, TRPM etc. Here, the abbreviation “TRP” is suitable.

Response: Reviewer is right. TRPC has been corrected and now it stands for transient receptor potential canonical.  

Lines 677-679: Michaelis et al (Glia, 2015, 63: 652) showed that knockout of STIM1/2 and Orai1 abolishes microglial SOCE and reduces phagocytosis and migration.

Response: Reference has been added and discussed in the text, as follows (new lines 703-706): “Blocking SOCE -via knocking down or knocking out STIM1/2 and Orai- reduces immune functioning, including phagocytosis, migration and cytokine production in primary isolated murine microglia”

Figures: The font size within the figures could be more consistent. For instance, a very big font is used for “PLASMA MEMBRANE” and “ASTROCYTES”, but “Na+” and “PLC” are tiny.

Response: Font size has been adjusted in both figures.

Reviewer 3 Report

Summary: In this article Rodriguez et al. review the role of Ca2+ dysregulation as an early event in Alzheimer’s disease (AD). The authors review the hypothesis that Ca2+ dysregulation in AD is caused by the hyperactivation or disruption of Ca2+ channels in the plasma membrane and intracellular compartments such as ER, mitochondria, and lysosome. This review is interesting and provides relevant insights to understand mechanisms of Ca2+ dysregulation not only in neurons but in other brain cell types including, astrocyte and microglia. The topic and content are interesting. However, some concerns should be addressed before publication:

  1. Most of the references refer only to reviews and not the original work. Please cite recent or original research articles. Few examples are:
    1. Line 68, Ref 22 (Review): Neuronal Ca2+ dyshomeostasis seems to manifest early in AD progression prior to the development of histopathological markers or clinical symptoms.
    2. Line 194-195 (Review): Ref 60, 61, 62
    3. Line 369 (Review): Ref 107
    4. Line 378-379, Ref.109, cite original work on MCU, De Stefani, Nature, 2011
    5. Line 382, Ref. 110, cite original work on MCUb, Raffaello, EMBO, 2015 and Lambert, Circulation, 2019
    6. Line 385 mention function of MICU1 as a gatekeeper and cooperative activator of the channel and cite Perocchi, Nature, 2010, Mallilankaraman, Cell, 2012, Csordas, Cell Metabolism 2013.

  1. Many sentences lack references. For example.
    1. Line 73-76, there are only two types of Food and Drug Administration (FDA) approved therapies for AD treatment –acetylcholinesterase inhibitors and N75 methyl-D-aspartate receptor (NMDAR) antagonists– and neither can cure or reverse the disease, but can, at least, transiently relieve patients’ symptoms
    2. Line 83-86, Ca2+ dyshomeostasis is an early molecular defect in AD and might precede Aβ and tau deposition…
    3. Line 93-95 - In the extracellular space, Ca2+ concentration is maintained between 1.1 - 1.4 mM, whereas resting cytosolic levels within neurons are maintained in the nM range (50 – 300 nM). After cell activation, intracellular Ca2+ concentrations increase rapidly to the μM range…..
    4. Line196- 201, Led by immunotherapy approaches, the efforts to target tau with therapeutics focus on reducing tau pathology by limiting the spread of extracellular tau across brain regions….
    5. Line 108-114, Proper intracellular Ca2+ homeostasis is crucial for many neuronal functions. Disruption of this homeostasis….

  1. Line 91 fix the 2. 2 to 2. Neuronal Ca2+ as a therapeutic target in AD
  2. The figures are of low quality. Please provide high-resolution figure files.
  3. Conclusion and future directions are very general. The authors should provide their own perspectives and comment on how future research may advance this particular area of research.

Author Response

  1. Most of the references refer only to reviews and not the original work. Please cite recent or original research articles. Few examples are:
    1. Line 68, Ref 22 (Review): Neuronal Ca2+ dyshomeostasis seems to manifest early in AD progression prior to the development of histopathological markers or clinical symptoms.
    2. Line 194-195 (Review): Ref 60, 61, 62
    3. Line 369 (Review): Ref 107
    4. Line 378-379, Ref.109, cite original work on MCU, De Stefani, Nature, 2011
    5. Line 382, Ref. 110, cite original work on MCUb, Raffaello, EMBO, 2015 and Lambert, Circulation, 2019
    6. Line 385 mention function of MICU1 as a gatekeeper and cooperative activator of the channel and cite Perocchi, Nature, 2010, Mallilankaraman, Cell, 2012, Csordas, Cell Metabolism 2013.

Response: References have been changed for a more recent ones or the original research articles. Additional suggested references by the reviewer have been added. References regarding MCU have been corrected and the function of MICU1 as a gatekeeper has been added as follows (lines 399-410):

“MICU1 and MICU2 also act as gatekeepers of MCU, with MICU1 getting involved when the extramitochondrial Ca2+ concentration is high, activating the channel open state. At low concentrations, the main player seems to be MICU2, leading to minimal accumulation of Ca2+ within mitochondria, thus preventing mitochondrial Ca2+ overload at resting conditions. MICU3, a paralogue of MICU1 and MICU2, is mainly expressed in the CNS, and has been proposed to enhance mitochondrial Ca2+ uptake in neurons. In regulating the MCU pore the mitochondrial Ca2+ uniporter regulator 1 (MCUR1) also plays a role. It has been suggested as a necessary player in MCU-mediated mitochondrial Ca2+ uptake. The small Ca2+-binding mitochondrial carrier protein (SCaMC, also known as SLC25A23), seems to also participate in the mitochondrial Ca2+ uptake by interacting with MCU and M1CU1”.

  1. Many sentences lack references. For example.
    1. Line 73-76, there are only two types of Food and Drug Administration (FDA) approved therapies for AD treatment –acetylcholinesterase inhibitors and N75 methyl-D-aspartate receptor (NMDAR) antagonists– and neither can cure or reverse the disease, but can, at least, transiently relieve patients’ symptoms
    2. Line 83-86, Ca2+ dyshomeostasis is an early molecular defect in AD and might precede Aβ and tau deposition…
    3. Line 93-95 - In the extracellular space, Ca2+ concentration is maintained between 1.1 - 1.4 mM, whereas resting cytosolic levels within neurons are maintained in the nM range (50 – 300 nM). After cell activation, intracellular Ca2+ concentrations increase rapidly to the μM range…..
    4. Line196- 201, Led by immunotherapy approaches, the efforts to target tau with therapeutics focus on reducing tau pathology by limiting the spread of extracellular tau across brain regions….
    5. Line 108-114, Proper intracellular Ca2+ homeostasis is crucial for many neuronal functions. Disruption of this homeostasis….

Response: References have been added to the sentences missing references according to reviewer suggestions.

  1. Line 91 fix the 2. 2 to 2. Neuronal Ca2+ as a therapeutic target in AD

Response: Numbers have been fixed.

  1. The figures are of low quality. Please provide high-resolution figure files

Response: Figures have been additionally provided in PDF for a better quality.

  1. Conclusion and future directions are very general. The authors should provide their own perspectives and comment on how future research may advance this particular area of research.

Response: New comments have been added to future directions, including:

(lines 768-773) “Current FDA approved AD treatments target plasma Ca2+ channels, but more specific approaches are needed to target other prevalent and disrupted intracellular Ca2+ signaling pathways, such as those of the ER or mitochondria. As our knowledge in Ca2+ dysregulation in AD grows, it seems more obvious that targeting these other sources of Ca2+ dysregulation could be an effective therapeutic strategy”.

And

(lines 779-781) “Additionally, reversal of cellular processes that are disrupted by Aβ or tau accumulation - including Ca2+ dyshomeostasis - early diagnosis, and/or lifestyle changes would also be necessary for successful therapeutic intervention.